# Molecular Mechanisms of Tumor Progression and Novel Therapeutic and Diagnostic Strategies in Mesothelioma

**DOI:** 10.3390/ijms26094299

**Published:** 2025-05-01

**Authors:** Taketo Kato, Ichidai Tanaka, Heng Huang, Shoji Okado, Yoshito Imamura, Yuji Nomata, Hirofumi Takenaka, Hiroki Watanabe, Yuta Kawasumi, Keita Nakanishi, Yuka Kadomatsu, Harushi Ueno, Shota Nakamura, Tetsuya Mizuno, Toyofumi Fengshi Chen-Yoshikawa

**Affiliations:** 1Department of Thoracic Surgery, Nagoya University Graduate School of Medicine, Nagoya 466-8560, Japan; taketokato63@gmail.com (T.K.); huang.heng.t3@s.mail.nagoya-u.ac.jp (H.H.); s-okado@med.nagoya-u.ac.jp (S.O.); y-imamura@med.nagoya-u.ac.jp (Y.I.); yujinomata@yahoo.co.jp (Y.N.); takenaka.hirofumi.u1@f.mail.nagoya-u.ac.jp (H.T.); hwatanabe@med.nagoya-u.ac.jp (H.W.); ykawasumi@med.nagoya-u.ac.jp (Y.K.); knakanishi@med.nagoya-u.ac.jp (K.N.); ykadomatsu@med.nagoya-u.ac.jp (Y.K.); h-ueno@med.nagoya-u.ac.jp (H.U.); shota197065@med.nagoya-u.ac.jp (S.N.); te.mizuno@med.nagoya-u.ac.jp (T.M.); 2Department of Respiratory Medicine, Nagoya University Graduate School of Medicine, Nagoya 466-8560, Japan; tanaka.ichidai.a9@f.mail.nagoya-u.ac.jp

**Keywords:** mesothelioma, Hippo signaling pathway, *BAP1*, FAK, *OXTR*, *CHST4*, immune checkpoint inhibitors

## Abstract

Mesothelioma is characterized by the inactivation of tumor suppressor genes, with frequent mutations in neurofibromin 2 (*NF2*), BRCA1-associated protein 1 (*BAP1*), and cyclin-dependent kinase inhibitor 2A (*CDKN2A*). These mutations lead to disruptions in the Hippo signaling pathway and histone methylation, thereby promoting tumor growth. *NF2* mutations result in Merlin deficiency, leading to uncontrolled cell proliferation, whereas *BAP1* mutations impair chromatin remodeling and hinder DNA damage repair. Emerging molecular targets in mesothelioma include mesothelin (*MSLN*), oxytocin receptor (*OXTR*), protein arginine methyltransferase (*PRMT5*), and carbohydrate sulfotransferase 4 (*CHST4*). *MSLN*-based therapies, such as antibody–drug conjugates and immunotoxins, have shown efficacy in clinical trials. *OXTR*, upregulated in mesothelioma, is correlated with poor prognosis and represents a novel therapeutic target. *PRMT5* inhibition is being explored in tumors with *MTAP* deletions, commonly co-occurring with *CDKN2A* loss. *CHST4* expression is associated with improved prognosis, potentially influencing tumor immunity. Immune checkpoint inhibitors targeting PD-1/PD-L1 have shown promise in some cases; however, resistance mechanisms remain a challenge. Advances in multi-omics approaches have improved our understanding of mesothelioma pathogenesis. Future research will aim to identify novel therapeutic targets and personalized treatment strategies, particularly in the context of epigenetic therapy and combination immunotherapy.

## 1. Introduction

Mesothelioma is a universally lethal and highly aggressive cancer that mainly affects the pleura; it is increasing in incidence worldwide and has a well-established causal relationship with asbestos exposure [1,2,3]. Despite improvements in our understanding of its complex pathobiology, including extensive alterations in genetics, epigenetics, the tumor microenvironment, and immunobiology, platinum–antifolate combination chemotherapy (cisplatin and pemetrexed) remains the treatment of choice. This regimen, which is used both for unresectable cases and as part of multimodal therapy for resectable lesions, has shown a tumor response rate of approximately 45.5%, progression-free survival (PFS) of 6.1 months, and overall survival (OS) of between 8.0 and 13.3 months, with a very poor 5-year survival rate of only 5–10% [4,5]. Surgical intervention, as evidenced by the MARS 2 trial, has not conferred a considerable survival benefit and is associated with a higher incidence of grade ≥ 3 adverse events than chemotherapy alone [6]. Considering these limitations, the recent exponential growth in mesothelioma research has prompted the development of novel therapeutic strategies. Extensive genomic analyses have revealed frequent inactivation of key tumor suppressors, such as neurofibromin 2 (*NF2*), BRCA1-associated protein 1 (*BAP1*), and cyclin-dependent kinase inhibitor 2A (*CDKN2A*), through various mechanisms, thereby providing new targets for treatment [7]. Among emerging therapies, immunotherapy has shown particular promise; phase III trials have reported improved survival with agents such as nivolumab and ipilimumab compared with conventional chemotherapy, and neoadjuvant immunotherapy trials have further suggested potential benefits [8,9,10,11,12]. With the expansion of immunotherapeutic options, the identification of reliable biomarkers for predicting prognosis and treatment response will become critical for the optimization of future therapeutic strategies. In this review, we explore the latest advancements in our understanding of mesothelioma genetics, epigenetics, the tumor microenvironment, and immunobiology. We examine how these findings translate into clinical applications and highlight emerging therapeutic strategies under development.

## 2. Frequent Inactivation of Tumor Suppressor Genes in Mesothelioma

### 2.1. Genetic Landscape and Clinical Implications of Interpatient Heterogeneity

Chronic inflammation plays a pivotal role in the development of mesothelioma caused by asbestos and other mineral fibers. At the sites where these fibers are deposited, mesothelial cells often undergo programmed necrosis, which leads to the release of high mobility group box 1 (HMGB1). HMGB1 recruits macrophages and activates inflammasome pathways, thereby triggering the release of inflammatory cytokines and the activation of NF-κB signaling. This proinflammatory environment facilitates mesothelial cell survival, proliferation, and accumulation of mutations, ultimately contributing to the development of mesothelioma [13].

Extensive studies of the genetic landscape of mesothelioma have reported that the most frequent mutations involve the inactivation of tumor suppressor genes, which are critical regulators of cell growth and division [14,15,16]. These mutations occur through various mechanisms, including single nucleotide variants, copy number losses, gene fusions, and splicing alterations. The most commonly inactivated tumor suppressor genes in mesothelioma are *NF2*, *BAP1*, and *CDKN2A*. Comprehensive analyses of the mesothelioma genome revealed that key genetic alterations often disrupt pathways involved in the Hippo signaling pathway and histone methylation, both of which are essential in regulating cell proliferation, growth, and survival [16].

Despite the relatively low incidence of point mutations in genes directly associated with cancer, mesothelioma is characterized by substantial global inactivation of tumor suppressor genes via CpG methylation, a process that silences genes by adding a methyl group to cytosine nucleotides in DNA [17]. This widespread methylation serves as a key feature that distinguishes between mesothelioma and normal pleural cells. Understanding these genetic abnormalities is crucial for developing targeted therapies that inhibit tumor progression and improve patient outcomes.

One of the defining characteristics of mesothelioma is its high level of morphological and genetic heterogeneity across patients, which is reflected in its various histological subtypes. The most common subtypes are epithelioid and sarcomatoid, with the latter typically associated with poor prognosis [18]. The epithelioid subtype is generally less aggressive, whereas the sarcomatoid subtype often presents with a more invasive and rapidly progressing form of the disease. The biphasic subtype, which contains epithelioid and sarcomatoid components, also tends to have a poorer prognosis than purely epithelioid tumors [19].

Recent genomic studies have highlighted distinct genetic alterations between these subtypes, which may explain the differences in their clinical behavior [16]. For example, mutations in *TP53*, a gene frequently mutated across numerous cancers, are relatively rare in epithelioid mesotheliomas but are common in the biphasic and sarcomatoid subtypes, albeit still at a frequency of <10%. This suggests that *TP53* plays a role in the increased aggressiveness observed in these subtypes [16]. Despite significant advancements in our understanding of the genomic differences between mesothelioma subtypes, there remains a critical gap in knowledge regarding which mutations act as clonal or subclonal drivers of the disease.

### 2.2. NF2 Inactivation and Hippo Signaling Pathway

The *NF2* gene, located on chromosome 22q, is frequently mutated or deleted in mesothelioma [14,20]. Merlin, which is encoded by the *NF2* gene, exerts its tumor-suppressive effects by controlling the expression of oncogenic genes through the activation of Hippo signaling (Figure 1). Merlin regulates contact inhibition and controls organ size by inhibiting the oncogenic transcriptional coactivators yes-associated protein 1 (YAP1) and transcriptional coactivator with PDZ-binding motif (TAZ). YAP1 encodes a transcription factor that regulates many tumorigenic genes, including survivin (also known as BIRC5), cellular inhibitor of apoptosis 1 (cIAP1; also known as *BIRC2*), KI67, and *MYC* [21]. *NF2* loss results in YAP1 activation, thereby driving tumorigenesis [22]. In addition, YAP1 amplification is commonly observed in primary mesothelioma tumors [23], further supporting the role of Hippo signaling pathway disruption in mesothelioma development. Furthermore, studies have demonstrated the frequent loss of large tumor suppressor kinase 1 (LATS1) and large tumor suppressor kinase 2 (LATS2), kinases that negatively regulate YAP1, contributing to the hyperactivation of the Hippo signaling pathway [24,25]. In relation to LATS2 and YAP, AJUBA, a binding partner of LATS2, negatively regulates YAP activity through the LATS family. AJUBA activation inhibits mesothelioma cell proliferation [26]. Together, *NF2* loss, YAP1 amplification, and LATS1/2 inactivation highlight the central role of this pathway in mesothelioma. Mice harboring heterozygous *Nf2* mutations exhibit increased susceptibility to asbestos-induced mesothelioma than mice with wild-type *Nf2* [27]. In these established tumors, mutational loss of the remaining wild-type *Nf2* allele has been reported, suggesting that asbestos directly promotes *Nf2* mutation [28]. Recently, VT3989, a TEAD inhibitor targeting the Hippo-YAP pathway, was well tolerated and demonstrated promising antitumor activity, including partial responses, in patients with mesothelioma and *NF2*-mutant tumors. The clinical benefit rate in mesothelioma was 57%, which support further development [29].

Preclinical studies have demonstrated that mesothelioma cells lacking *NF2* are sensitive to inhibitors of focal adhesion kinase (FAK), which disrupts the interaction of FAK with SRC and the p85 regulatory subunit of PI3K [30]. Merlin, the protein encoded by *NF2*, typically inhibits FAK by preventing its phosphorylation; thus, when *NF2* is lost, FAK activity increases, leading to enhanced cell invasiveness. The reintroduction of *NF2* into these cells reverses this invasive behavior, suggesting that FAK targeting is a potential therapeutic strategy for *NF2*-deficient mesotheliomas [31,32]. In phase I studies, the FAK inhibitor GSK2256098 was shown to improve PFS in patients with mesothelioma who had low Merlin expression [33]. However, the COMMAND trial, a phase II trial of the FAK inhibitor defactinib, recruited 344 patients with mesothelioma, and low levels of Merlin did not improve disease outcomes [34].

### 2.3. BAP1 Inactivation and Therapeutic Potential

Approximately 65% of mesothelioma cases involve the inactivation of *BAP1*, a tumor suppressor gene located on chromosome 3p21.1 [35,36]. While germline mutations in *BAP1* are rare, they markedly increase the risk of developing mesothelioma and other cancers. *BAP1* functions as a deubiquitinase, a protein that removes ubiquitin molecules from other proteins, and is involved in processes such as DNA double-strand break repair and epigenetic regulation [37,38]. *BAP1* loss disrupts these processes, making cells more vulnerable to DNA damage and other cancer-promoting factors. In mesothelioma, *BAP1* inactivation is associated with alterations in the polycomb repressive complex 2 (PRC2) pathway, which regulates histone methylation, a key process in gene expression control (Figure 2). This disruption suggests that mesotheliomas with BAP1 mutations are sensitive to inhibitors targeting EZH2, a component of the PRC2 complex [39]. A phase II clinical trial has yielded promising results for tazemetostat, an EZH2 inhibitor, in mesothelioma patients with *BAP1* inactivation, offering hope for targeted therapies in this patient subgroup. Further refinement of biomarkers for tazemetostat activity could help identify a subset of tumors most likely to derive prolonged benefit or shrinkage from this therapy [40].

### 2.4. CDKN2A Inactivation in Mesothelioma

The *CDKN2A* gene, located on chromosome 9p21, is the most frequently inactivated tumor suppressor gene in mesothelioma, with homozygous deletions occurring in 50–100% of cases [41,42]. This gene plays a pivotal role in regulating the cell cycle, and its loss is associated with more aggressive disease and reduced patient survival, particularly in cases of nonepithelioid mesothelioma [43,44]. In asbestos-induced mesotheliomas, *CDKN2A* mutations often occur after biallelic inactivation of *NF2*, another key tumor suppressor gene [45]. *CDKN2A* can also be epigenetically inactivated by methylation in approximately 19% of cases [46,47]. *CDKN2A* encodes two proteins, INK4A (p16) and ARF (p14 in humans), through alternative reading frames, a process by which different proteins are produced from the same gene (Figure 3) [48]. INK4A acts as an inhibitor of cyclin-dependent kinases CDK4 and CDK6, proteins involved in driving the cell cycle from the G1 to the S phase. INK4A loss removes this checkpoint, leading to unchecked cell proliferation [49]. Meanwhile, ARF interacts with the MDM2 protein, promoting its degradation and thus activating p53, a major tumor suppressor involved in the prevention of abnormal cell growth [50,51,52]. ARF loss results in increased MDM2 levels which, in turn, represses p53 activity, further contributing to tumorigenesis [53]. In addition, a fusion transcript involving YY1 and EWSR1 may indirectly suppress p53 by stabilizing the ARF–MDM2 interaction [54]. Efforts are underway to treat patients with p16-deficient mesothelioma with the CDK4/6 inhibitor abemaciclib. The MiST2 study was a single-arm, open-label, phase II clinical trial conducted at two centers in the UK. This trial reported disease control in 54% of patients within 12 weeks and is being investigated further in randomized trials as a targeted stratified therapy [55].

## 3. Other Targets in Mesothelioma

### 3.1. Mesothelin

Mesothelin (*MSLN*) is a 40-kDa glycoprotein, originally believed to be produced solely by mesothelioma cells; however, later investigations revealed its expression in numerous normal human tissues [56,57]. Approximately 77% of patients with epithelial mesothelioma tested positive for *MSLN*, a finding not observed in sarcomatoid tumors [58,59]. To measure MSLN levels, several blood assays that detect soluble MSLN-related peptides have been developed, among which MESOMARK [60] and the N-ERC/MSLN test [61] are the most prominent. Although these biomarkers alone are insufficient for diagnosing mesothelioma, researchers are optimistic that combining MSLN assays with tests for additional markers, such as calretinin, could enhance the detection accuracy [62].

Given the high expression of *MSLN* in epithelial mesothelioma, a range of innovative treatment strategies are currently being evaluated in various phases of clinical trials. These emerging therapies include the anti-MSLN immunotoxin SS1P, the monoclonal antibody amatuximab (MORAb-009) [63], and the antibody–drug conjugate anetumab ravtansine (BAY 94-9343) [64]. After a phase I study established the safety of SS1P, its combination with pemetrexed and cisplatin was tested in 24 chemotherapy-naïve patients with unrespectable mesothelioma, which led to an objective response rate (ORR) of 77% [65]. Amatuximab, a mouse–human chimeric monoclonal antibody that selectively binds to *MSLN*, mainly blocks the interaction between *MSLN* and the antigen CA125/MUC16 [66]. In a phase II trial, 89 patients with unresectable mesothelioma received a regimen combining amatuximab with pemetrexed and cisplatin. Although this approach did not improve the PFS of patients compared with controls, it yielded promising outcomes, with a median OS of 14.8 months and an ORR of 39% [63]. Anetumab ravtansine (BAY 94-9343) is a novel, highly potent, and selective antibody–drug conjugate composed of a human anti-MSLN antibody associated with the maytansinoid tubulin inhibitor DM4 through a disulfide-containing linker, with an average of three lysyl conjugations [67]. In vivo studies have shown that anetumab ravtansine specifically localizes to MSLN-positive tumors and effectively inhibits tumor growth in subcutaneous and orthotopic xenograft models [68]. Moreover, it can exert a bystander effect on adjacent tumor cells that do not express MSLN. Promising results were reported in a study of 148 patients with mesothelioma, ovarian cancer, and other malignancies, in which anetumab ravtansine targeted MSLN-expressing tumors [69]. Following these outcomes, a randomized trial comparing the combination of anetumab ravtansine and pembrolizumab with pembrolizumab alone was conducted; however, no significant differences were observed in response rates or progression-free survival between the two groups. Notably, high levels of soluble MSLN—which can neutralize anti-MSLN antibodies—were associated with poorer survival in patients receiving anetumab ravtansine [70].

### 3.2. Oxytocin Receptor

The oxytocin receptor (*OXTR*), a G-protein-coupled receptor, was recently significantly upregulated in mesothelioma cell lines compared with other cancer types. Cases with elevated *OXTR* expression have poor clinical outcomes [71]. Typically, *OXTR* is mainly expressed in tissues such as the mammary glands and uterine myometrium at the end of pregnancy, where it facilitates milk release and uterine contractions [72]. *OXTR* is also present in the central nervous system, influencing various behaviors [73]. Notably, *OXTR* expression surges in the uterine myometrium toward the end of pregnancy, although oxytocin secretion remains relatively stable [72]. It has been demonstrated that *OXTR* expression is regulated by interleukin (IL)-6 and IL-1β, but the addition of these cytokines to mesothelioma cell lines with low *OXTR* expression did not markedly increase *OXTR* mRNA levels [74]. Analysis of mesothelioma cell lines and The Cancer Genome Atlas (TCGA) data revealed a significant correlation between *OXTR* mRNA expression and *NF2* gene inactivation. The experimental results further confirmed that *NF2* knockdown increased *OXTR* expression, suggesting that *NF2* negatively regulates *OXTR* transcription [71]. Reduction of *OXTR* expression notably inhibited the proliferation of mesothelioma cell lines with high *OXTR* levels by disrupting the tumor cell cycle. Furthermore, OXTR antagonists reduced mesothelioma cell growth, and oral administration of the OXTR antagonist cligosiban hindered mesothelioma tumor progression. These findings point to the role of *OXTR* in mesothelioma cell proliferation and its potential as a therapeutic target.

### 3.3. PRMT5

The protein arginine methyltransferase 5 (*PRMT5*) is a methyltransferase enzyme that works in conjunction with its cofactor methylosome protein 50 and uses S-adenosyl methionine (SAM) as a methyl donor to symmetrically dimethylate arginine residues on target proteins [75,76]. This posttranslational modification plays a pivotal role in several cellular processes, such as RNA splicing, transcription, and translation. *PRMT5* has been identified as a synthetic lethal target in cancers with a homozygous deletion of the *MTAP* gene, which is commonly co-deleted with the tumor suppressor gene *CDKN2A* [77]. *MTAP* gene deletions are frequently observed in various cancers, including mesothelioma [16,78]. Typically, *MTAP* catalyzes the breakdown of 5′-deoxy-5′-methylthioadenosine (MTA), a metabolite that competes with SAM for binding to *PRMT5* and acts as a moderate inhibitor of the ability of *PRMT5* to add symmetric dimethylarginine (SDMA) modifications to its protein targets. *MTAP* loss leads to MTA accumulation, making *MTAP*-deficient cancer cells susceptible to *PRMT5* inhibition [79]. PRMT5 inhibitors exerted promising effects on in vitro models of mesothelioma with MTAP deletion, as its inhibition selectively inhibited tumor cell proliferation and downregulated genes related to cell cycle and epithelial–mesenchymal transition [80]. MRTX1719, developed using a structure-based drug design, is a small-molecule inhibitor that targets the PRMT5/MTA complex. The drug is currently being evaluated in a phase I/II clinical trial (NCT05245500) for its safety and effectiveness against advanced solid tumors with *MTAP* deletions. Early findings from this trial have indicated six confirmed objective responses, suggesting that MRTX1719 can effectively inhibit SDMA production at well-tolerated doses, leading to substantial tumor shrinkage [81].

### 3.4. CHST4

Recently, the MARS 2 trial demonstrated that surgery did not offer all patients significant survival advantages compared with chemotherapy alone, although this trial has been criticized in various aspects [6]. Consequently, there is a growing demand for biomarkers that can predict postoperative outcomes and help tailor therapeutic strategies for each patient with mesothelioma. Through an extensive survival analysis of the TCGA mesothelioma dataset, the carbohydrate sulfotransferase 4 (*CHST4*) gene emerged as a promising indicator of favorable OS in patients with mesothelioma [82]. In this study, enrichment analysis of genes associated with favorable prognosis, including *CHST4*, revealed immune-related ontological terms, whereas analysis of genes associated with unfavorable prognosis highlighted cell-cycle-related terms. Notably, *CHST4* mRNA expression in mesothelioma demonstrated a significant correlation with immune–gene signatures representing tumor-infiltrating lymphocytes [83]. As a sulfotransferase expressed in high endothelial venules [84], *CHST4* plays a pivotal role in the immune system by facilitating the homing of lymphocytes to lymphoid organs (Figure 4) [85]. It is hypothesized that *CHST4* contributes to tumor immunity against malignant tumors. Further functional analysis of this gene may advance the development of immunotherapeutic strategies.

To explore the relationship between *CHST4* expression and patient outcomes, researchers examined *CHST4* protein levels via immunohistochemistry in 23 surgical samples from patients with mesothelioma who underwent complete macroscopic resection. The researchers quantified *CHST4* expression using a scoring system that incorporated the percentage of stained cells and staining intensity. The analysis revealed that patients with higher *CHST4* scores experienced significantly better postoperative outcomes, with a median OS of 107.8 months compared with 38.0 months in those with lower scores [82]. Although pleural thickness is now reflected in the ninth edition of the TNM classification [86] and has been reported as a risk factor for postoperative complications [87], it was not correlated with the intensity of CHST4 expression, suggesting that CHST4 may serve as an independent prognostic marker.

## 4. Effect of Immunotherapy on Mesothelioma

### 4.1. Tumor Microenvironment in Mesothelioma

In mesothelioma, the tumor microenvironment (TME) is a key factor influencing tumor development, immune suppression, and treatment resistance [88]. Chronic inflammation caused by asbestos exposure shapes a distinctive TME populated by immunosuppressive elements, such as tumor-associated macrophages (TAMs), regulatory T cells, cancer-associated fibroblasts, and dysfunctional T cells. Among these, M2-like TAMs are particularly prevalent and promote tumor progression through the release of inhibitory cytokines, such as IL-10 and TGF-β, correlating with unfavorable clinical outcomes [89,90]. The immunosuppressive landscape is further reinforced by the adenosine signaling pathway, which dampens immune cell activity and supports TAM proliferation [91]. Notably, the composition and immune features of the TME vary across different mesothelioma histological subtypes. Nonepithelioid subtypes, such as sarcomatoid and biphasic mesothelioma, are generally associated with a more immunosuppressive microenvironment, characterized by high expression of immune checkpoint molecules, including PD-L1, PD-L2, TIM-3, and CTLA-4 [92]. Overall, the interplay of stromal and immune cells in the TME significantly contributes to the aggressiveness and treatment resistance of mesothelioma, highlighting the importance of developing treatment strategies that particularly target the TME [93].

### 4.2. Chemoimmunotherapy

Recent advances in immune checkpoint inhibition, particularly targeting molecules such as cytotoxic T lymphocyte-associated protein 4 (*CTLA4*), programmed cell death 1 (PD-1), and programmed death ligand 1 (PD-L1), have proven successful in enhancing immune responses and T cell function across various malignancies [94]. Based on encouraging outcomes in patients with relapsed disease [8], the combination therapy of nivolumab and ipilimumab, which specifically target PD-1 and *CTLA4*, respectively, has shown superior efficacy compared to standard chemotherapy as a first-line treatment (median survival: 18.1 vs. 14.1 months), particularly among patients with nonepithelioid mesothelioma (CheckMate 743) [11]. Consequently, frontline treatment with ipilimumab and nivolumab is strongly recommended for patients with biphasic or sarcomatoid mesothelioma. This predefined interim analysis supported the US FDA’s approval of this combination therapy in 2020, marking it as the first FDA-approved systemic treatment for mesothelioma since 2004 [9]. The impact of this regimen on older or frailer patients as compared with clinical trial participants remains an area of interest.

Table 1 presents recent clinical trials of first-line and salvage immunotherapy conducted for mesothelioma. To date, many phase III trials on maintenance therapy—long-term treatment aimed at delaying relapse after induction chemotherapy—have shown considerable improvements in OS [34,95,96]. However, in certain cases, rechallenging with platinum–pemetrexed chemotherapy has shown potential benefits [97]. In addition, the combination of immune checkpoint inhibitors (ICIs) and chemotherapy has exerted a synergistic effect and is currently recognized as a standard of care for non-small cell lung cancer [98]. KEYNOTE-028, pembrolizumab therapy for previously treated patients with PD-L1-positive, also showed well-tolerated findings [99]. Consequently, randomized trials are currently evaluating the efficacy of PD-1 or PD-L1 inhibitors in combination with chemotherapy for mesothelioma, with promising results reported in the Canadian Cancer Trials Group (NCT02784171) [100]. In addition, phase II trials, such as PrECOG0505 (NCT02899195) [101] and DREAM [102], have laid the groundwork for the DREAM3R trial, an international phase III study that compared standard cisplatin–pemetrexed therapy with and without durvalumab, a PD-L1 inhibitor [103].

Vascular endothelial growth factor (VEGF) plays a key role in mesothelioma by promoting tumor angiogenesis and immune suppression. It supports tumor growth and aids the tumor in evading immune responses by recruiting regulatory immune cells and increasing PD-L1 expression. VEGF targeting may inhibit tumor progression and immune evasion [104]. Findings from the Mesothelioma Avastin Cisplatin–Pemetrexed Study (MAPS) indicate that antiangiogenic agents enhance immune cell differentiation and activity [105], providing the rationale for the phase III BEAT-meso trial (NCT03762018), which is assessing the addition of the PD-L1 inhibitor atezolizumab to bevacizumab and standard chemotherapy (ABC) [106]. In this study, a notable improvement in median PFS for ABC was observed, which led to a numerical increase in median OS, although it was not statistically significant. In the prespecified histology-based analysis, ABC showed superior OS and PFS in nonepithelioid cases compared with BC.

Among recent salvage immunotherapy trials, the Japanese MERIT trial holds historical importance as the first to obtain regulatory approval for a salvage checkpoint inhibitor [107]. In contrast, the DETERMINE trial, which investigated tremelimumab monotherapy, revealed that single-agent *CTLA-4* blockade showed no benefit for mesothelioma [95]. Similarly, the PROMISE-Meso trial failed to demonstrate a survival advantage for pembrolizumab over investigator-selected chemotherapy (gemcitabine or vinorelbine) in PD-L1–positive patients [96]. However, the CONFIRM trial demonstrated that nivolumab markedly improved PFS and OS compared with placebo [108]. The administration of adenovirus-mediated interferon alfa-2b therapy is currently under evaluation in the phase III INFINITE trial, following promising phase II data showing a disease control rate of 88% [109]. This trial involves previously treated patients randomly assigned to receive intrapleural adenovirus therapy followed by either celecoxib and gemcitabine or celecoxib and gemcitabine alone, with treatment continuing until disease progression or treatment termination due to toxicity. These findings suggest that a subset of mesothelioma patients derives substantial benefits from immunotherapy. Until predictive biomarkers are further refined, all mesothelioma patients are recommended to receive an immunotherapy regimen at some stage of their treatment.

### 4.3. Cellular Therapy

Immune system modulation provides potential for more sustained disease control compared with chemotherapy and results in fewer toxic side effects. Researchers have investigated innovative approaches to altering immune system behavior. One such strategy involves genetically engineered T cells, known as chimeric antigen receptor T (CAR-T) cells, designed to specifically target MSLN—an antigen found mainly, although not exclusively, in mesothelioma cells [110,111,112]. A phase I clinical trial that investigated the intrapleural administration of these CAR-T cells in combination with an ICI to 18 patients with mesothelioma reported a median OS of 23.9 months, which is an exciting finding in such an early-phase study [113].

## 5. Conclusions

Recent advances in mesothelioma treatment—ranging from innovative medical devices to novel immunotherapy combinations—have injected renewed momentum into this field. However, despite these promising developments, improvements in patient survival remain modest, owing to the limited number of large-scale randomized trials and considerable interpatient heterogeneity. Accurate identification of histological subtypes is paramount for tailoring frontline systemic therapy, whereas ongoing clinical investigations continue to explore expanded options for patients with unresectable diseases. Notably, challenges persist in patients refractory to current immunotherapeutic approaches. To overcome these hurdles, future research must prioritize well-controlled, multicenter trials, the adoption of individualized treatment strategies, and the development of master protocols that facilitate global collaboration. Furthermore, the increasing incidence of mesothelioma in developing countries highlights the urgent need for effective, accessible, and scalable treatment solutions. Ultimately, a multidisciplinary approach that integrates molecular insights with clinical innovation is necessary to translate these advances into meaningful improvements in patient outcomes.

## Figures and Tables

**Figure 1 ijms-26-04299-f001:**
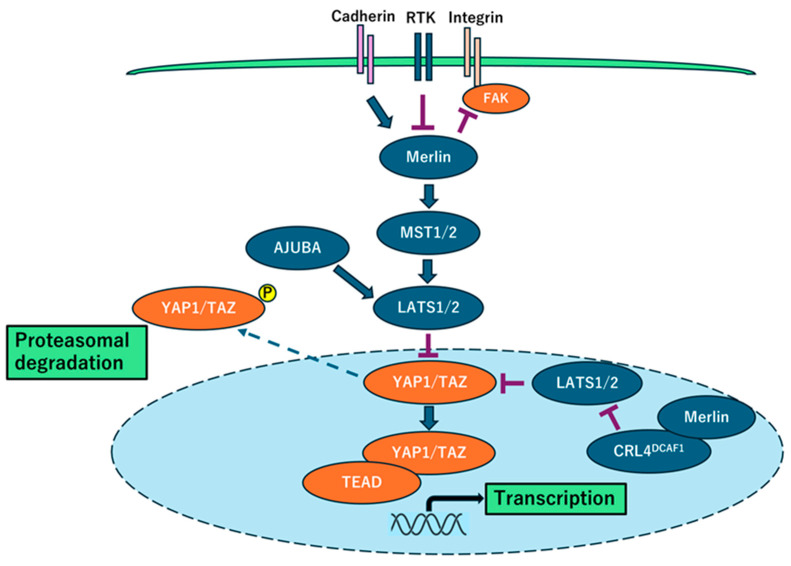
A schematic illustration depicting the dysregulation of the *NF2*/Merlin-Hippo signaling cascade in mesothelioma cells. Extracellular signals are transmitted through cell–cell interactions (via cadherins), cell–matrix interactions (via integrins), and growth factor receptors, such as receptor tyrosine kinases, influencing the tumor-suppressive function of Merlin. When Merlin is in an active, underphosphorylated state, it modulates the Hippo signaling cascade and inhibits the activity of YAP1/TAZ transcriptional coactivators. However, in mesothelioma cells, frequent alterations in Merlin (the *NF2* gene product) and key Hippo signaling pathway components, including LATS1/2, lead to YAP1/TAZ activation (underphosphorylation). This, in turn, promotes the expression of multiple pro-oncogenic genes.

**Figure 2 ijms-26-04299-f002:**
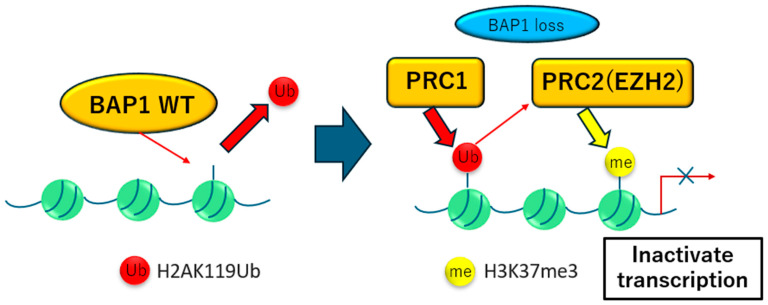
Transcriptional regulation machinery comprising BAP1, PRC1, and PRC2. BAP1 is a major deubiquitinase that removes H2AK119Ub, a histone marker deposited by the PRC1 complex. BAP1 loss leads to H2AK119Ub accumulation, which in turn promotes recruitment of the PRC2 complex including EZH2, resulting in elevated H3K27me3 levels and transcriptional repression.

**Figure 3 ijms-26-04299-f003:**
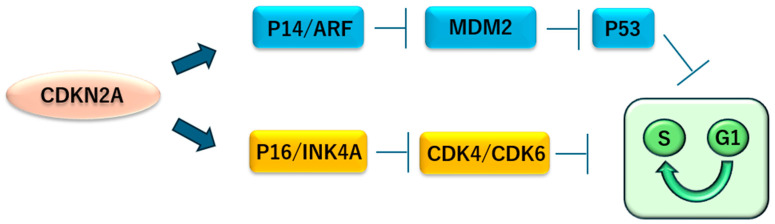
The CDKN2A gene produces two distinct tumor suppressor proteins through alternative reading frames, namely, p16INK4A (P16), derived from the alpha transcript, and p14ARF (P14), derived from the beta transcript. These proteins play pivotal roles in halting cell cycle progression, particularly during the G1 to S phase transition.

**Figure 4 ijms-26-04299-f004:**
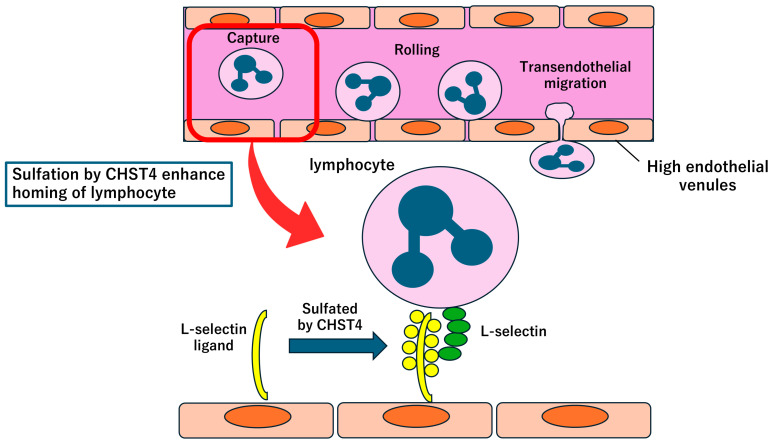
Role of *CHST4* in lymphocyte homing and mesothelioma. Lymphocytes begin their interaction with high endothelial venules by binding L-selectin to its ligand, leading to integrin-mediated adhesion and transendothelial migration. Sulfation by *CHST4* plays a pivotal role in the enhancement of lymphocyte homing and the chemokine-induced activation step of this sequential adhesion process. In mesothelioma, *CHST4* mRNA expression exhibited a substantial correlation with immune–gene signatures representing tumor-infiltrating lymphocytes. *CHST4* is hypothesized to contribute to tumor immunity against mesothelioma via lymphocyte infiltration.

**Table 1 ijms-26-04299-t001:** Recent clinical trials of first-line and salvage immunotherapy conducted in mesothelioma.

Trial	ID	Treatment	Status/Result
DREAM3R	NCT04334759	Randomized phase III trial of durvalumab with chemotherapy as first-line treatment	Active but not recruiting
MAPS	NCT00651456	Randomized phase III trial of bevacizumab plus standard chemotherapy as first-line treatment	Positive
BEAT-meso trial	NCT03762018	Randomized phase III trial comparing atezolizumab plus bevacizumab and standard chemotherapy with bevacizumab and standard chemotherapy as first-line treatment	Negative
PROMISE-Meso	NCT02991482	Randomized phase III trial comparing pembrolizumab with standard chemotherapy for pretreated mesothelioma	Negative
CONFIRM	NCT03063450	Placebo controlled phase III trial to evaluate the efficacy of nivolumab in relapsed mesothelioma	Positive
INFINITE	NCT03710876	Randomized phase III trial of intrapleural administration of adenovirus-delivered interferon alfa-2b in combination with celecoxib and gemcitabine for pretreated mesothelioma	Active but not recruiting
CheckMate 743	NCT0289929	Randomized phase III trial of nivolumab plus ipilimumab vs. pemetrexed and platinum as first-line treatment	Positive
KEYNOTE-028	NCT02054806	Single-arm phase Ib trial of pembrolizumab for previous treated mesothelioma	OS 18 months
DREAM	ACTRN 12616001170415	Single-arm phase II trial of durvalumab in combination with cisplatin and pemetrexed as first-line treatment	OS 18.4 months
PrECOG0505	NCT02899195	Sigle-arm phase II trial of durvalumab in combination with cisplatin and pemetrexed as the first-line treatment	OS 20.5 months

## Data Availability

Not applicable.

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
