# Peer review of "Molecular Mechanisms of Tumor Progression and Novel Therapeutic and Diagnostic Strategies in Mesothelioma"

_ijms, 2025, doi:10.3390/ijms26094299_

Round 1
Reviewer 1 Report
Comments and Suggestions for Authors
This is an appropriate and excellent review of the subject. A few suggestions are made in the hope of improving the manuscript.
1) In addition to Table 1, a Table of completed immunotherapy and chemoimmunotherapy trials would be useful. This would briefly include Keynote 028, Dream, Checkmate 743, PreCOG among others. This would complement the existing table of ongoing trials.
2) In the section on YAP pathway, ongoing studies with inhibitors such at VT3989 should be discussed. There are oral presentations from AACR (Cancer Res (2023) 83 (8_Supplement): CT006.) detailing very promising results.
3) I strongly suggest the authors add a section on VEGF targeted therapies for mesothelioma, as bevacizumab is an approved agent for this indication in North America. These trials would be MAPS, Beat-MESO, and ARCS-M for starters. Although it is clear that VEGF targeting has been surpassed by immunotherapy in treating mesothelioma, these studies are still important and offer a therapeutic option.
Author Response
We thank you for your comments, which have helped us to improve the manuscript. We have corrected the issues pointed out in the manuscript.
Comment 1: In addition to Table 1, a Table of completed immunotherapy and chemoimmunotherapy trials would be useful. This would briefly include Keynote 028, Dream, Checkmate 743, PreCOG among others. This would complement the existing table of ongoing trials.
Reply: Thank you for your valuable suggestions. We added KEYNOTE-028, Dream, Checkmate 743, PreCOG0505 in Table 1 and description about KEYNOTE-028 in Line 376-78 as below (Dream, Checkmate 743, PreCOG0505 were already mentioned in main manuscript).
KEYNOTE-028, pembrolizumab therapy for previously treated patients with PD-L1-positive, also showed well-tolerated findings [99].
Comment 2: In the section on YAP pathway, ongoing studies with inhibitors such at VT3989 should be discussed. There are oral presentations from AACR (Cancer Res (2023) 83 (8_Supplement): CT006.) detailing very promising results.
Reply: As following your suggestions, we added the description about above study in Line 135-138 as below.
Recently, VT3989, a TEAD inhibitor targeting the Hippo-YAP pathway, was well tolerated and demonstrated promising antitumor activity, including partial responses, in patients with mesothelioma and NF2-mutant tumors. The clinical benefit rate in mesothelioma was 57%, which support further development [29].
Comment 3: I strongly suggest the authors add a section on VEGF targeted therapies for mesothelioma, as bevacizumab is an approved agent for this indication in North America. These trials would be MAPS, Beat-MESO, and ARCS-M for starters. Although it is clear that VEGF targeting has been surpassed by immunotherapy in treating mesothelioma, these studies are still important and offer a therapeutic option.
Reply: Thank you for your suggestion. MAPS and Beat-MESO were already described in the manuscript. Therefore, we added about rationale of VEGF therapy in Line 385-389 as below and made an independent paragraph. About ARCS-M, it is a therapy targeting mesothelin, and it was included in the chapter of mesothelin (reference no. 62).
Vascular endothelial growth factor (VEGF) plays a key role in mesothelioma by promoting tumor angiogenesis and immune suppression. It supports tumor growth and aids the tumor in evading immune responses by recruiting regulatory immune cells and increasing PD-L1 expression. VEGF targeting may inhibit tumor progression and immune evasion [104].
Reviewer 2 Report
Comments and Suggestions for Authors
This manuscript is of interest and the content is consistent with the objectives of the Review.
Specific comments:
1) Line 60: “In this review, we explored the latest advancements in our understanding of mesothelioma genetics, epigenetics, tumor microenvironment, and immunobiology. We examined how these findings translate into clinical applications and highlight emerging therapeutic strategies under development.” However, the part regarding the tumor microenvironment in mesothelioma is almost completely missing in this review. Please, see the review by Cersosimo et al. (PMID: 34830817) which provides a comprehensive overview of immunity and the microenvironment in mesothelioma.
2) Lines 86–89: it is recommended to add a reference.
3) Figure 2 as currently presented, illustrates a generic process of lymphocyte homing in peripheral lymph nodes. However, given that the review focuses on mesothelioma and its molecular mechanisms, the representation appears decontextualized with respect to the focus of the manuscript. It is suggested to reword the figure caption to explicitly clarify the potential role of CHST4 in the context of mesothelioma, integrating a brief explanation of its specific impact on tumor immunity in this type of neoplasia. Alternatively, it is possible to consider the insertion of a complementary image or the modification of the current scheme to make it more relevant to the topic discussed.
4) Previous paper of Barbarino et al. (PMID: 32301278), exploring the effect of PRMT5 silencing in MTAP-deleted mesothelial tumor, suggesting a promising therapeutic strategy should be included.
Comments on the Quality of English LanguageThe English should be revised to more clearly express the research
Author Response
We thank you for your comments, which have helped us to improve the manuscript. We have corrected the issues pointed out in the manuscript. The manuscript has been carefully reviewed again by an experienced editor whose first language is English.
Comment 1: Line 60: “In this review, we explored the latest advancements in our understanding of mesothelioma genetics, epigenetics, tumor microenvironment, and immunobiology. We examined how these findings translate into clinical applications and highlight emerging therapeutic strategies under development.” However, the part regarding the tumor microenvironment in mesothelioma is almost completely missing in this review. Please, see the review by Cersosimo et al. (PMID: 34830817) which provides a comprehensive overview of immunity and the microenvironment in mesothelioma.
Reply: Thank you for your suggestions. We made one paragraph and described the tumor microenvironment in mesothelioma in line 336-352 as below.
In mesothelioma, the tumor microenvironment (TME) is a key factor influencing tumor development, immune suppression, and treatment resistance [88]. Chronic inflammation caused by asbestos exposure shapes a distinctive TME populated by immunosuppressive elements, such as tumor-associated macrophages (TAMs), regulatory T cells, cancer-associated fibroblasts, and dysfunctional T cells. Among these, M2-like TAMs are particularly prevalent and promote tumor progression through the release of inhibitory cytokines, such as IL-10 and TGF-β, correlating with unfavorable clinical outcomes [89,90]. The immunosuppressive landscape is further reinforced by the adenosine signaling pathway, which dampens immune cell activity and supports TAM proliferation [91]. Notably, the composition and immune features of the TME vary across different mesothelioma histological subtypes. Nonepithelioid subtypes, such as sarcomatoid and biphasic mesothelioma, are generally associated with a more immunosuppressive microenvironment, characterized by high expression of immune checkpoint molecules, including PD-L1, PD-L2, TIM-3, and CTLA-4 [92]. Overall, the interplay of stromal and immune cells in the TME significantly contributes to the aggressiveness and treatment resistance of mesothelioma, highlighting the importance of developing treatment strategies that particularly target the TME [93].
Comment 2: Lines 86–89: it is recommended to add a reference.
Reply: As following your suggestion, we also added reference no. 19 for the description in Line 86-89 (currently Line 98-102).
Comment 3: Figure 2 as currently presented, illustrates a generic process of lymphocyte homing in peripheral lymph nodes. However, given that the review focuses on mesothelioma and its molecular mechanisms, the representation appears decontextualized with respect to the focus of the manuscript. It is suggested to reword the figure caption to explicitly clarify the potential role of CHST4 in the context of mesothelioma, integrating a brief explanation of its specific impact on tumor immunity in this type of neoplasia. Alternatively, it is possible to consider the insertion of a complementary image or the modification of the current scheme to make it more relevant to the topic discussed.
Reply: Thank you for your valuable suggestions. As followed your opinion, we modified caption for Figure 4. It includes the CHST4 role not only for lymphocyte homing also for mesothelioma as below.
Figure 4. Role of CHST4 in lymphocyte homing and mesothelioma. Lymphocytes begin their interaction with high endothelial venules by binding L-selectin to its ligand, leading to integrin-mediated adhesion and transendothelial migration. Sulfation by CHST4 plays a pivotal role in the enhancement of lymphocyte homing and the chemokine-induced activation step of this sequential adhesion process. In mesothelioma, CHST4 mRNA expression exhibited a substantial correlation with immune-gene signatures representing tumor-infiltrating lymphocytes. CHST4 is hypothesized to contribute to tumor immunity against mesothelioma via lymphocyte infiltration.
Comment 4: Previous paper of Barbarino et al. (PMID: 32301278), exploring the effect of PRMT5 silencing in MTAP-deleted mesothelial tumor, suggesting a promising therapeutic strategy should be included.
Reply: We described PRMT5 inhibition for in vitro model of mesothelioma and added suggested paper as a reference in Line 285-288 as below.
PRMT5 inhibitors exerted promising effects on in vitro models of mesothelioma with MTAP deletion as its inhibition selectively inhibited tumor cell proliferation and downregulated genes related to cell cycle and epithelial–mesenchymal transition [80]
Round 2
Reviewer 2 Report
Comments and Suggestions for Authors
The revised version of the manuscript is sufficiently improved